# Combined Multimorbidity and Polypharmacy Patterns in the Elderly: A Cross-Sectional Study in Primary Health Care

**DOI:** 10.3390/ijerph18179216

**Published:** 2021-09-01

**Authors:** Grant Stafford, Noemí Villén, Albert Roso-Llorach, Amelia Troncoso-Mariño, Mònica Monteagudo, Concepción Violán

**Affiliations:** 1Programa de Máster en Salud Pública, Universitat Pompeu Fabra, 08003 Barcelona, Spain; grant.stafford95@gmail.com; 2Unitat Transversal de Recerca (UTR), Fundació Institut Universitari per a la Recerca a l’Atenció Primària de Salut Jordi Gol i Gurina (IDIAPJGol), 08007 Barcelona, Spain; aroso@idiapjgol.org (A.R.-L.); mmonteagudo@idiapjgol.org (M.M.); 3Àrea del Medicament i Servei de Farmàcia, Atenció Primària Barcelona Ciutat, Institut Català de la Salut (ICS), 08015 Barcelona, Spain; nvillenr.bcn.ics@gencat.cat (N.V.); atroncoso@gencat.cat (A.T.-M.); 4Programa de Doctorat en Metodologia de la Recerca Biomèdica i Salut Pública, Universitat Autònoma de Barcelona, Bellaterra (Cerdanyola del Vallès), 08193 Barcelona, Spain; 5Universitat Autònoma de Barcelona, Bellaterra (Cerdanyola del Vallès), 08193 Barcelona, Spain; 6Department of Clinical Sciences, University of Barcelona and IDIBELL, L’Hospitalet de Llobregat, 08907 Barcelona, Spain; 7Unitat de Suport a la Recerca Metropolitana Nord, Fundació Institut Universitaria per a la Recerca a l’Atenció Primària de Salut Jordi Gol i Gurina (IDIAPJGol), 08303 Mataró, Spain

**Keywords:** multimorbidity, polypharmacy, elderly, primary healthcare, chronic disease, clustering, combined patterns, machine learning

## Abstract

(1) Background: The acquisition of multiple chronic diseases, known as multimorbidity, is common in the elderly population, and it is often treated with the simultaneous consumption of several prescription drugs, known as polypharmacy. These two concepts are inherently related and cause an undue burden on the individual. The aim of this study was to identify combined multimorbidity and polypharmacy patterns for the elderly population in Catalonia. (2) Methods: A cross-sectional study using electronic health records from 2012 was conducted. A mapping process was performed linking chronic disease categories to the drug categories indicated for their treatment. A soft clustering technique was then carried out on the final mapped categories. (3) Results: 916,619 individuals were included, with 93.1% meeting the authors’ criteria for multimorbidity and 49.9% for polypharmacy. A seven-cluster solution was identified: one non-specific (Cluster 1) and six specific, corresponding to diabetes (Cluster 2), neurological and musculoskeletal, female dominant (Clusters 3 and 4) and cardiovascular, cerebrovascular and renal diseases (Clusters 5 and 6), and multi-system diseases (Cluster 7). (4) Conclusions: This study utilized a mapping process combined with a soft clustering technique to determine combined patterns of multimorbidity and polypharmacy in the elderly population, identifying overrepresentation in six of the seven clusters with chronic disease and chronic disease-drug categories. These results could be applied to clinical practice guidelines in order to better attend to patient needs. This study can serve as the foundation for future longitudinal regarding relationships between multimorbidity and polypharmacy.

## 1. Introduction

The global life expectancy at birth has increased from 52.6 years in 1960 to 72.6 years in 2018 [1]. While it is certain that people are living longer on average, this does not necessarily mean they are living healthier lives, as an increase in life expectancy anticipates an increase in morbidity [2,3]. As individuals age, the body changes and experiences a state of physical decline, resulting in weaker defenses and easier acquisition of chronic illnesses in the later years of life [2,4]. The diagnosis of two or more chronic diseases in the same individual is referred to as multimorbidity [5].

Multimorbid individuals tend to be prescribed a high number of medications in order to combat their diagnosed chronic illnesses. Consumption of prescribed drugs holds a higher prevalence and relevance in older adults, and complications could include potentially inappropriate prescribing [6]. While a homogeneous operational definitional is lacking throughout the field, literature supports the definition of polypharmacy as the consumption of five or more drugs daily in the same individual [7]. Polypharmacy is considered a critical public health problem that is related to drug-drug and drug-disease interactions, adverse drug events [8,9,10,11,12], falls, hospital admissions and mortality [13,14]. Polypharmacy has been on the rise over the past several decades [11] and is highly associated to multimorbidity [8].

In a world with an ageing population, the burdens of multimorbidity and polypharmacy have undue individual and system-wide impacts on health. While there exists a growing amount of literature regarding multimorbidity and polypharmacy, the vast majority of studies analyze polypharmacy as descriptive drugs in multimorbidity patterns, focus almost exclusively on one topic or the other without meaningfully connecting the two, or examine the disease rather than the individual as the unit of analysis [15,16,17]. Furthermore, medication is considered a proxy variable to disease [15,18,19], and, for this reason, jointly analyzing multimorbidity and polypharmacy can produce an overestimation error due to the fact that people with prevalent diseases such as diabetes or cardiovascular diseases are treated with many medications for both clinical conditions and risk factors and, for this reason, are overestimated. To avoid this, drug groups can be analyzed according to their associated disease, thereby preventing prevalent diseases from being overestimated. This type of approach would permit a better understanding of the patient groups and, at the same time, facilitate strategies aimed at prevention, diagnosis, and treatment because it includes diseases with or without drug treatment.

As far as we understand, little research has been completed regarding methods that simultaneously analyze the combined patterns of multimorbidity and polypharmacy at an individual level. Machine-learning soft clustering models are a robust tool capable of performing such an analysis. Cluster analysis involves assigning individuals to a certain cluster so that the items (i.e., units of analysis—diseases and drugs) are as similar as possible, while individuals in different clusters are as least similar as possible. Cluster identification is based on similarity measures, and their choice is reliant upon the data and/or the reason for analysis [20]. Hard clustering forces each individual to belong to only one cluster, while soft clustering (also called fuzzy clustering) grants varying degrees of membership, thus allowing for the individual to pertain to multiple clusters [20]. The aim of this study was to determine combined patterns of multimorbidity and polypharmacy in the Catalan population 65–99 years of age through a machine-learning soft clustering technique that incorporates the research team’s mapping of chronic disease and drug associations.

## 2. Materials and Methods

### 2.1. Setting, Design, and Population

Catalonia, an autonomous community of Spain, is a Mediterranean region with 7,515,398 reported inhabitants for the year 2012 [21]. Universal health coverage is established for residents in Spain by the National Health Service and is implemented in a decentralized fashion through each of the seventeen autonomous communities [22]. In Catalonia, the Catalan Health Institute (CHI) manages over 283 primary care centers, offering health services to over six million residents [23].

A cross-sectional study was performed on the baseline year (2012) of a longitudinal study (2012–2016) using electronic health records (EHRs) in Catalonia. Inclusion criteria for the cross-sectional study population allowed for individuals 65–99 years of age on 31 December 2011, who survived until 31 December 2012, and had at least one visit to a CHI-managed primary care center during the longitudinal study period (2012–2016). No new entries were permitted in the study, and dropouts were due to either death or transfer to another primary care center outside of CHI governance. A total of 916,619 eligible individuals were included at the baseline year (Figure 1).

### 2.2. Data Source

The Information System for Research in Primary Care (SIDIAP) contains EHRs from the primary care centers managed by the CHI [24]. The SIDIAP database, in addition to clinical information, contains demographic, laboratory, and invoiced drug information, with every datapoint linked to the individual via an anonymous and unique personal identifier.

### 2.3. Variables

The SIDIAP database was the single source of information for all variables. All variables were analyzed exclusively within the study period (2012).

#### 2.3.1. Chronic Diseases and Multimorbidity

All diseases in the SIDIAP database were coded according to the International Classification of Diseases, Version 10 (ICD-10). An operational definition for multimorbidity was based on the 60 chronic disease categories determined by Calderón-Larrañaga et al. in the Swedish National study of Aging and Care in Kungsholmen (SNAC-K) [25]. Each chronic disease category was included as an individual binary variable, and multimorbidity was defined via a dichotomous variable as the presence of two or more diagnoses from the 60 chronic disease categories. However, only chronic disease categories with ≥2% prevalence in the study population were included for final analysis, thus leaving 47 SNAC-K chronic disease categories in total (Appendix B Table A1).

#### 2.3.2. Drugs and Classification

Invoiced drugs recorded in the SIDIAP database included drugs dispensed in pharmacies. Drugs received in hospital and/or dispensed by a hospital pharmacy and all other drugs not subsidized through the national health system were excluded from this study. Drugs were coded according to the Anatomical Therapeutic Chemical (ATC) Classification System, which categorizes drugs through various levels of specificity into groups (hereafter referred to as “drug categories”) according to the targeted organ/system and their chemical, pharmacological, and therapeutic properties [26]. Drug categories with ≥1% prevalence in the study population were included for final analysis. Chronic use for invoiced drugs was determined for individuals who were invoiced three or more packages of the same drug categories during the study period. While not meeting the prevalence or chronic use criteria, the drug category *Other Drugs Affecting Bone Structure and Mineralization* (ATC 4th level code “M05BX”) was added to the study due to its twice-a-year treatment regimen for chronic diseases. Each drug category was included as an individual binary variable, and polypharmacy was defined via a dichotomous variable as chronic use in the same individual for five or more different drug categories (ATC 4th level) from the eighty-nine drug categories outlined in the following section.

#### 2.3.3. Grouping of Drugs and Mapping to Chronic Diseases

The research team identified 89 different drug categories (ATC 4th level) (Appendix B Table A2) associated to the 60 chronic disease categories mentioned prior via a thorough revision of several databases that are well-known international clinical guidelines for each disease [27,28,29]. A mapping was done of all 60 SNAC-K chronic disease categories [25] and 89 drug categories. Drug categories were mapped to the SNAC-K chronic disease categories for which they are prescribed to treat. Chronic disease-drug categories were then created in the form of dichotomous variables for individuals who were diagnosed with a SNAC-K chronic disease category and, depending on disease management criteria, invoiced 0, 1, or 1+ of the mapped drug categories. (For example, the Allergy disease category was mapped with seven drug categories. An individual would qualify in this category if diagnosed with a disease pertaining to the Allergy disease category and invoiced at least one of the seven drug categories). A total of seven of the final 47 SNAC-K chronic diseases categories require non-pharmacological treatments or treatment with drugs excluded from this study and, therefore, could not be mapped. The remaining 40 SNAC-K chronic disease categories were mapped to drug categories, resulting in 29 chronic disease-drug categories containing ≥ 2% prevalence in the study population (see Appendix B Table A3 for mapping example; see Appendix A for complete mapping process). The seven chronic disease categories requiring non-pharmacological treatment or treatment with drugs excluded from this study and the 29 chronic disease-drug categories, all with prevalence ≥ 2%, were included to determine combined patterns of multimorbidity and polypharmacy (Figure 2).

#### 2.3.4. Other Variables

Pertinent demographic data analyzed in the study includes age (measured in years), sex (female or male), and socioeconomic status (measured by the MEDEA [Mortality in Spanish Small Areas and Socioeconomic and Environmental Inequalities] Index via quintiles from least deprived to most deprived for urban areas, while rural areas were sorted into an independent category [30]).

### 2.4. Statistical Analysis

Descriptive statistics were applied to summarize preliminary findings.

Due to the high dimensionality of the SIDIAP database, dimension-reduction techniques were exercised through PCA Mix, an application of principal component analysis (PCA) for numeric original variables and multiple correspondence analysis (MCA) for binary variables. This method reduced the size of the database while maintaining the complexity of the original data. The Karlis–Saporta–Spinaki rule was applied in order to select the appropriate number of dimensions to preserve [31].

Using the reduced database, the combined multimorbidity and polypharmacy patterns were determined through a fuzzy c-means (FCM) clustering algorithm [32], incorporating the twenty-nine chronic disease-drug categories and the seven chronic disease categories with non-pharmacological treatment or treatment with drugs excluded from this study, all of which satisfied the prevalence threshold of ≥2% in the study population. To obtain a range with the ideal number of clusters, validation indices [33] were calculated (Appendix A). The outcome of the FCM clustering algorithm was a determined number of models with different numbers of clusters in each model, as calculated by the validation indices. Each model contained varying degrees of disease-medication association, and the final model of clusters was determined by the research team according to clinical relevance.

The clusters were described in two parts: (1) observed/expected ratios (O/E ratios) were calculated by dividing the prevalence of a chronic disease or chronic disease-drug category in a specific cluster by the prevalence of the same chronic disease or chronic disease-drug category in the entire study population; (2) exclusivity was determined by dividing the number of individuals with a chronic disease or chronic disease-drug category in a specific cluster by the amount of all individuals with the same chronic disease or chronic disease-drug category in the entire study population. A threshold of two for the O/E ratio was set in order for a disease/medication to be considered a relevant part of a cluster [34,35]. An exclusivity threshold of 30% was a secondary, but not determining, factor when evaluating the chronic disease or chronic disease-drug categories association with a cluster.

All analyses were performed in R version 4.0.3 and Stata version 15. Specifically, R was used to run the PCA mix and FCM clustering algorithm; Stata was used for data management.

## 3. Results

Of the 916,619 eligible individuals 65 years and over (women: 57.8%; mean age: 75.4; standard deviation; 7.4), 853,085 (93.1%) satisfied the criteria for multimorbidity, and 457,576 (49.9%) for polypharmacy (Figure 3). The most frequent chronic disease categories in the population were hypertension (71%), dyslipidemia (50.9%), osteoarthritis and other degenerative joint diseases (32.8%), obesity (28.7%), and diabetes (25.1%) [Appendix B Table A1], with a median of six chronic diseases (interquartile range [IQR] 4.0–8.0) per person. The most prevalent drug categories included proton pump inhibitors (44.3%), HMG CoA reductase inhibitors (38.2%), anilides (28.4%), platelet aggregation inhibitors, excluding heparin (35.6%), and benzodiazepine derivatives (20.9%) [Appendix B Table A2], with a median of 5 drugs (IQR 2.0–8.0) per person.

The authors identified a seven-cluster solution for combined patterns of multimorbidity and polypharmacy. Cluster 2 to Cluster 7 contained over 99% multimorbidity in each cluster and reported higher overrepresentation values (O/E ratio > 2) for at least one of the chronic disease or chronic disease-drug categories. Characteristics of the study participants in each cluster are detailed in Appendix B Table A4. Principal results and the most frequent chronic disease and chronic disease-drug categories by cluster (Table 1) are highlighted below:

Cluster 1 (non-specific) included a substantial number of individuals that do not present any overrepresented chronic disease or chronic disease-drug category (O/E ratios are below two and exclusivity values are below 30%). Cluster 1 was also the cluster with the lowest average age (74.20 years, SD 7.47) and the lowest percentage of individuals with multimorbidity (81.74%) and polypharmacy (15.38%).

Diabetes (Cluster 2): The only category that surpassed the O/E ratio threshold was chronic disease-drug group for “diabetes” (O/E ratio 2.15), with exclusivity of 41.93%.

Neurological and musculoskeletal, female dominant (Cluster 3): 69.05% of the individuals in this cluster were female, with exclusivity for the chronic disease-drug groups for “peripheral neuropathy”, “dorsopathies”, and “other musculoskeletal and joint diseases” all surpassing 30%. Similar to Cluster 3 is behavioral, neurological, and musculoskeletal, female dominant (Cluster 4), which also has a high proportion of females (75.08%) and contains many of the chronic disease-drug groups from the neurological and musculoskeletal categories. However, the differentiating factor is the emphasis on behavioral chronic disease-drug groups, with several groups pertaining to this category as well.

Cardio-cerebrovascular and renal (Cluster 5): The vast majority (six out of eight) of the significant groups in this cluster are cardiovascular related, with exclusivity of chronic disease-drug groups for “peripheral vascular disease” and “ischemic heart disease” both exceeding 30%. This cluster contained the highest proportion of polypharmacy (83.65%).

Cardiovascular, renal, inflammatory, and respiratory (Cluster 6): Cardiovascular categories occupy the top three spots, with chronic disease-drug categories for “atrial fibrillation” (O/E ratio 5.99) and “heart failure” (O/E ratio 5.70), and chronic disease category “bradycardias and conduction disorders” (O/E ratio 4.14), all exceeding an exclusivity of 30%. This cluster contained the highest proportions for individuals diagnosed with 10 or more chronic diseases (30.82%), individuals prescribed 10 or more chronic drugs (27.71%), and individuals with 10 or more primary care visits (75.38%). Polypharmacy was nearly that of Cluster 5 (83.54%).

The multisystem pattern identified in Cluster 7 represents the smallest group of individuals from the study population (5.41%) and contains several overrepresented diseases corresponding to multiple systems, in which individuals with digestive disease dominate the cluster.

Socioeconomic status (measured by MEDEA index) between categories appears to remain relatively stable in Cluster 2 to Cluster 7, with most deprived fluctuating between 13.63–15.79% and least deprived between 14.90–16.57%. Those in rural health settings fluctuate between 19.34–23.83% in Cluster 2 to Cluster 7.

## 4. Discussion

### 4.1. Key Results

In this article, patterns for multimorbidity and polypharmacy were analyzed in a joint manner by means of creating a variable relating each drug to one or more diseases for which they are indicated. Such patterns were obtained via specified criteria that allowed for the identification of multimorbid individuals that also qualified as polymedicated (those who were invoiced more than five distinct drugs in one year). This study ultimately identified distinct patterns with singular characteristics, with some more profound in women than in men. This is a key point in order to carry out a stricter follow up of the individuals who have a higher risk of presenting secondary effects caused by drugs or drug interaction.

Cluster 1 (non-specific) did not overrepresent any disease and Cluster 7 (multisystemic) overrepresented diseases from many systems are less specific than Clusters 2 to 6. These cluster types have been identified in other studies [35,36]. In our study, we were able to identify overrepresented diseases that do not have any prescribed drug in primary healthcare, such as neoplastic disease, deafness and hearing loss, and bradycardias and conduction diseases. These data suggest that diseases and drugs should be considered in combination to more efficiently identify multimorbidity patterns [37].

It was observed that the chronic disease-drug categories “hypertension” and “dyslipidemia” are distributed throughout the seven obtained clusters. Individuals with these common pathologies are therefore not grouped together in any one specific cluster; rather, they are spread throughout all seven. The fuzzy c-means method is a technique that classifies individuals based on cluster probability membership, and, given that there were many individuals with these chronic diseases and associated drugs in the study, the results are homogenously distributed throughout the study population [37].

The patterns from Cluster 2 to Cluster 6 group individuals who share similar chronic disease-drug categories. Cluster 2 is classified as diabetes because 41.93% of individuals included in the cluster are diabetic. Although not reaching the O/E threshold of two, associations are still observed between diabetes and obesity, cataracts, and neoplasms, all diseases that are frequently presented in diabetic individuals [38,39,40].

Cluster 3 (neurological and musculoskeletal) and Cluster 4 (behavioral, neurological, and musculoskeletal) include patterns predominantly in women. The clustering model identified these two groups of individuals that present similar health problems with singular characteristics. The patterns within behavioral, neurological, and musculoskeletal show an association with autoimmune diseases that are also more frequent in women [41].

Cluster 5 (cardio-cerebrovascular and renal) and Cluster 6 (cardiovascular, renal, inflammatory, and respiratory) include chronic disease-drug patterns that primarily treat cardiac, cardiovascular, cerebrovascular, and renal pathologies. This association, which has been observed in a previous study that indicated a pattern of increased mortality [16], concerns closely related pathologies that also share risk factors and even treatments. This study allows for the grouping of individuals with similar chronic disease and chronic disease-drug categories into two differentiated and specific clusters, given that Cluster 6 (cardiovascular, renal, inflammatory, and respiratory) includes a chronic disease-drug category for inflammatory pathologies as well as a chronic disease-drug group for respiratory diseases. Chronic inflammation as a mechanism in atherosclerosis carries a higher risk for cardiovascular and cerebrovascular events [42]. Autoimmune diseases and arthrosis have also been linked to an increase in cardiovascular disease, although this could be due to adverse effects of the treatments for these diseases, such as corticoids [43,44].

A small group of individuals who presented polypharmacy did not satisfy the condition for multimorbidity. This could be due to the individuals having received multiple treatments during the study period without them coinciding. On the other hand, there is also the possibility that some pathologies could be treated with five or more drugs simultaneously for certain patients, with neuropathy being a prime example.

The inclusion of the same study variable for chronic diseases and drugs indicated for their respective diseases’ treatment is an advantage that allows for a better identification of the individuals and, thereby, allows to better orient the clinical management of these groups of individuals.

The methodology used for mapping drugs and chronic diseases permitted the identification of a new chronic diseases-drug category. This new variable applied to fuzzy clustering methods is less susceptible to outliers in the data, choice of distance measure, and the inclusion of inappropriate or irrelevant variables [35,45,46].

This mapping process between chronic diseases and drugs, together with the application of the fuzzy method, will be very useful for multimorbidity studies where drugs are used as a proxy variable.

Soft clustering methods offer a new methodological approach towards understanding the relationships between specific diseases or drugs in individuals. This is an essential step in improving the care of patients and the quality of health systems. Analyzing multimorbidity patterns based on drugs permits the identification of patient subgroups with different clinical approaches and attention. Our analysis focuses on groups of patients with specific diseases and drugs as opposed to other studies centered in diseases.

This methodology can be applied in a large variety of studies using electronic health records to study polypharmacy and multimorbidity.

### 4.2. Comparison to the Literature

Studies performed with clustering methods concerning multimorbidity and polypharmacy are novel but becoming more frequent [47]; however, these studies generally approach multimorbidity with a polypharmacy perspective or vice versa. To the authors’ knowledge, this study is the first in conveying a combined approach to multimorbidity and polypharmacy by combining both topics into one inclusive variable to determine joint outcomes. Although, as previously mentioned, some of the patterns obtained (neurological and musculoskeletal, female dominant; behavioral, neurological, and musculoskeletal, female dominant; and cardio-cerebrovascular and renal) coincide with the literature [17,37,48].

### 4.3. Strengths and Limitations

The sample size, with nearly one million individuals, is a clear advantage of the study. Considering individuals as the primary variable for analysis rather than diseases [34], the study employs a large, high-quality database composed of primary health care records representative of the Catalan population aged ≥ 65 years [24], an age group that is more susceptible to present health problems related to drugs. The inclusion of all drugs that are indicated for different diseases with a minimum number of packages facilitates the identification of those individuals who should be more thoroughly monitored. Regarding the method, soft clustering offers a methodologic focus in understanding the relations between specific diseases and individuals. The analysis of multimorbidity and polypharmacy patterns can identify subgroups of patients with different associated diseases and drugs. The extensive mapping process of drug categories to their respective SNAC-K chronic disease categories provides a further level of detail than most other clustering techniques within the literature. This can ultimately serve as a framework for future studies that wish to map diseases and drugs under certain conditions to determine combined patterns.

Some limitations of this study should be considered. Individuals who met initial selection criteria but sought care outside of CHI governance were ineligible, possibly introducing selection bias for individuals who chose to seek care in a private healthcare facility. However, this is a small group of the population, and the results of this study can be applied to the general population. Drugs were recorded via invoices, and hospital drugs were not included, which could have influenced the calculation for polypharmacy. Concerning the clusters, the results obtained in this study are similar to those obtained in other studies without prior mapping; nevertheless, we consider that this methodology more precisely determines which pathologies and drugs are overrepresented in each cluster, thus more adequately defining patient profiles. Finally, while the fuzzy c-means clustering technique used in this study is an unsupervised, exploratory method, the authors believe that, combined with a vigorous internal validation system, this technique produces robust results and minimizes potential pitfalls.

## 5. Conclusions

This study utilized a mapping process combined with a soft clustering technique to determine combined patterns of multimorbidity and polypharmacy in the elderly Catalan population in 2012, identifying overrepresentation in 6 of the 7 clusters with chronic disease and chronic disease-drug categories. Cluster 2 to Cluster 6 provided recognizable patterns, predominantly in diabetes; neurological and musculoskeletal, female dominant and behavioral, neurological, and musculoskeletal, female dominant; and cardio-cerebrovascular and renal and cardiovascular, renal, inflammatory, and respiratory. These patterns further highlight the differences between sexes, specifically within neurological and musculoskeletal, female dominant and behavioral, neurological, and musculoskeletal, female dominant.

The combined patterns of multimorbidity and polypharmacy identified in this study will contribute key information to the evaluation of multimorbid and polymedicated individuals, facilitating the identification of these subgroups of individuals that require specific attention. The obtained results could be applied to clinical practice guidelines differentiating distinct population groups, such as multimorbid individuals with and without associated diseases and/or polypharmacy. Due to the relationship between multimorbidity and polypharmacy over long periods of time, this analysis could serve as the base to deepen this relationship in further longitudinal studies. The patterns obtained in this research will allow for an in-depth study of the prescription of multiple medications in elderly people in relation to medication-related problems.

## Figures and Tables

**Figure 1 ijerph-18-09216-f001:**
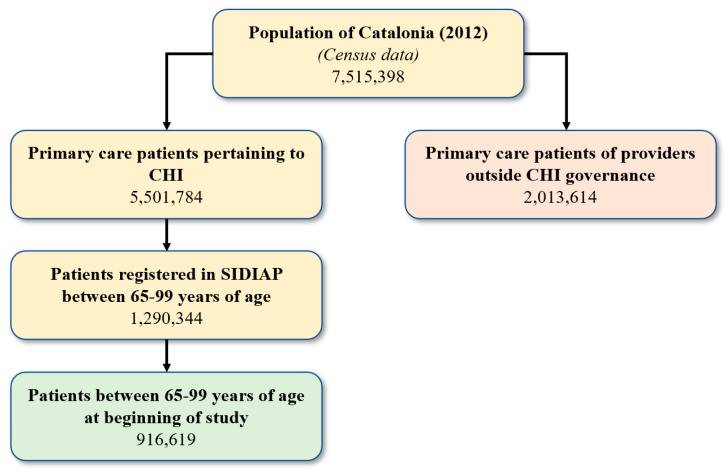
Estimated population study according to selection criteria.

**Figure 2 ijerph-18-09216-f002:**
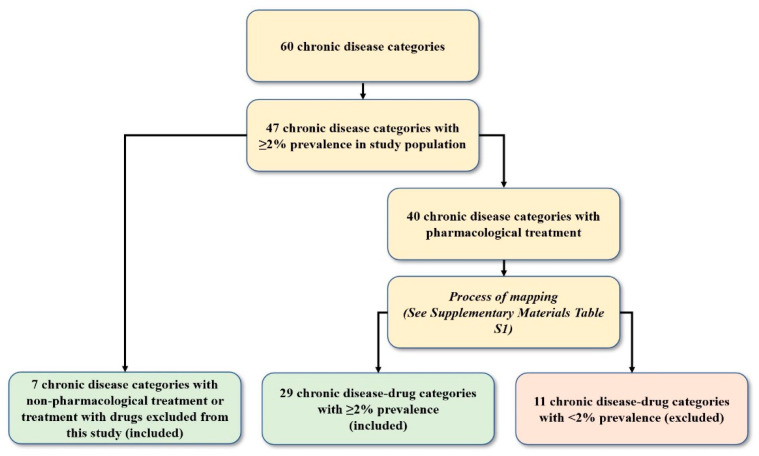
Number of significant chronic disease or chronic disease-drug categories after mapping.

**Figure 3 ijerph-18-09216-f003:**
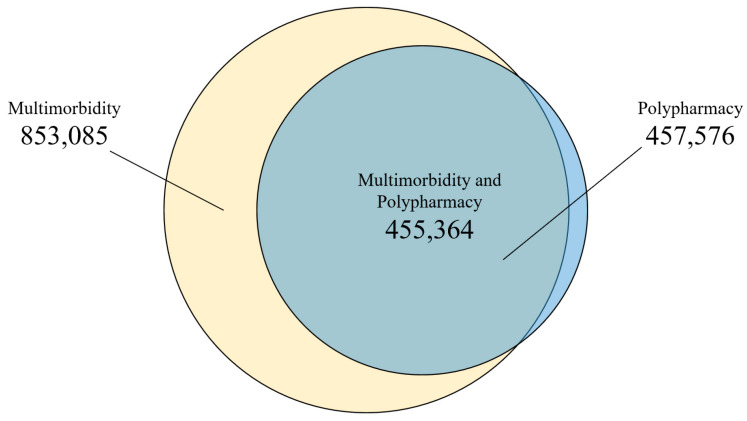
Multimorbid and polymedicated individuals in the study aged 65–99 years (*n* = 916,619, Catalonia, 2012).

**Table 1 ijerph-18-09216-t001:** Most frequent 15 chronic disease or chronic disease-drug categories in individuals aged 65–99 years by cluster (*n* = 916,619, Catalonia, 2012).

Pattern	Disease or Disease-Medication Category	O	O/E Ratio	EX
1 Non-Specific (*n* = 344,958: 37.63%)	Chronic disease group for solid neoplasms	12.86	0.86	32.37
	Chronic disease-drug group for prostate diseases	7.82	0.77	29.11
	Chronic disease-drug group for osteoporosis	7.74	0.74	27.74
	Chronic disease group for deafness and hearing loss	5.91	0.60	22.58
	Chronic disease-drug group for COPD, emphysema, and chronic bronchitis	4.26	0.53	20.03
	Chronic disease-drug group for esophagus, stomach, and duodenum diseases	3.77	0.52	19.54
	Chronic disease-drug group for thyroid diseases	2.84	0.51	19.34
	Chronic disease group for cataract and lens diseases	8.47	0.50	18.65
	Chronic disease-drug group for dementia	1.56	0.47	17.61
	Chronic disease-drug group for hypertension	26.41	0.46	17.48
	Chronic disease group for bradycardias and conduction diseases	1.12	0.40	15.13
	Chronic disease-drug group for sleep disorders	2.22	0.39	14.64
	Chronic disease group for obesity	11.29	0.39	14.82
	Chronic disease-drug group for dyslipidemia	12.32	0.38	14.40
	Chronic disease group for chronic pancreas, biliary tract, and gallbladder diseases	1.09	0.37	13.78
2 Diabetes (*n* = 178,457: 19.47%)	**Chronic disease-drug group for diabetes**	**39.52**	**2.15**	**41.93**
	Chronic disease-drug group for glaucoma	10.75	1.78	34.65
	Chronic disease group for obesity	49.61	1.73	33.68
	Chronic disease-drug group for dyslipidemia	55.12	1.71	33.33
	Chronic disease-drug group for hypertension	84.37	1.48	28.89
	Chronic disease-drug group for thyroid diseases	7.44	1.34	26.17
	Chronic disease-drug group for chronic kidney diseases	13.92	1.32	25.69
	Chronic disease-drug group for ischemic heart disease	8.52	1.11	21.61
	Chronic disease-drug group for cerebrovascular diseases	6.54	1.01	19.69
	Chronic disease group for cataract and lens diseases	17.15	1.00	19.54
	Chronic disease-drug group for peripheral vascular disease	2.55	1.00	19.47
	Chronic disease-drug group for prostate diseases	9.58	0.95	18.46
	Chronic disease group for solid neoplasms	14.06	0.94	18.31
	Chronic disease group for deafness and hearing loss	8.42	0.85	16.64
	Chronic disease group for chronic pancreas, biliary tract, and gallbladder diseases	2.40	0.80	15.67
3 Neurological and Musculoskeletal, Female Dominant (*n* = 102,750: 11.21%)	**Chronic disease-drug group for peripheral neuropathy**	**9.73**	**3.08**	**34.56**
	**Chronic disease-drug group for dorsopathies**	**24.05**	**2.88**	**32.24**
	**Chronic disease-drug group for other musculoskeletal and joint diseases**	**22.42**	**2.73**	**30.64**
	**Chronic disease-drug group for other genitourinary diseases**	**8.68**	**2.44**	**27.32**
	**Chronic disease-drug group for glaucoma**	**14.68**	**2.43**	**27.25**
	**Chronic disease-drug group for osteoarthritis and other degenerative joint diseases**	**41.87**	**2.16**	**24.19**
	**Chronic disease group for deafness and hearing loss**	**19.68**	**2.00**	**22.40**
	Chronic disease-drug group for neurotic, stress-related, and somatoform diseases	20.38	1.99	22.36
	Chronic disease group for cataract and lens diseases	33.75	1.98	22.14
	Chronic disease-drug group for depression and mood diseases	23.16	1.86	20.90
	Chronic disease-drug group for osteoporosis	18.91	1.80	20.17
	Chronic disease-drug group for colitis and related diseases	18.22	1.79	20.09
	Chronic disease-drug group for sleep disorders	9.93	1.74	19.51
	Chronic disease-drug group for other psychiatric and behavioral diseases	3.18	1.58	17.71
	Chronic disease-drug group for esophagus, stomach, and duodenum diseases	11.29	1.55	17.41
4 Behavioral, Neurological, and Musculoskeletal, Female Dominant (*n* = 90,287: 9.85%)	**Chronic disease-drug group for other psychiatric and behavioral diseases**	**7.42**	**3.69**	**36.34**
	**Chronic disease-drug group for neurotic, stress-related, and somatoform diseases**	**37.33**	**3.65**	**36.00**
	**Chronic disease-drug group for peripheral neuropathy**	**10.93**	**3.46**	**34.09**
	**Chronic disease-drug group for depression and mood diseases**	**41.75**	**3.36**	**33.10**
	**Chronic disease-drug group for dorsopathies**	**27.22**	**3.25**	**32.06**
	**Chronic disease-drug group for other musculoskeletal and joint diseases**	**26.65**	**3.25**	**32.00**
	**Chronic disease-drug group for other genitourinary diseases**	**9.81**	**2.76**	**27.14**
	**Chronic disease-drug group for sleep disorders**	**15.07**	**2.64**	**26.02**
	**Chronic disease-drug group for osteoarthritis and other degenerative joint diseases**	**47.21**	**2.43**	**23.97**
	**Chronic disease-drug group for colitis and related diseases**	**22.46**	**2.21**	**21.76**
	Chronic disease-drug group for osteoporosis	20.89	1.99	19.58
	Chronic disease-drug group for esophagus, stomach, and duodenum diseases	13.80	1.90	18.70
	Chronic disease-drug group for thyroid diseases	8.76	1.58	15.60
	Chronic disease-drug group for autoimmune diseases	3.12	1.43	14.13
	Chronic disease group for deafness and hearing loss	13.77	1.40	13.77
5 Cardio-cerebrovascular and Renal (*n* = 80,855: 8.82%)	**Chronic disease-drug group for peripheral vascular disease**	**12.02**	**4.71**	**41.57**
	**Chronic disease-drug group for ischemic heart disease**	**29.85**	**3.89**	**34.32**
	**Chronic disease-drug group for cerebrovascular diseases**	**19.34**	**2.99**	**26.37**
	**Chronic disease-drug group for heart failure**	**21.10**	**2.83**	**24.94**
	**Chronic disease group for bradycardias and conduction diseases**	**7.04**	**2.53**	**22.33**
	**Chronic disease-drug group for atrial fibrillation**	**15.98**	**2.39**	**21.06**
	**Chronic disease-drug group for other psychiatric and behavioral diseases**	**4.63**	**2.30**	**20.33**
	**Chronic disease-drug group for chronic kidney diseases**	**21.85**	**2.07**	**18.27**
	Chronic disease-drug group for COPD, emphysema, chronic bronchitis	15.85	1.98	17.45
	Chronic disease-drug group for colitis and related diseases	19.64	1.93	17.04
	Chronic disease-drug group for anemia	11.32	1.93	17.00
	Chronic disease-drug group for neurotic, stress-related, and somatoform diseases	18.28	1.79	15.79
	Chronic disease-drug group for prostate diseases	17.96	1.78	15.67
	Chronic disease-drug group for depression and mood diseases	21.75	1.75	15.45
	Chronic disease-drug group for sleep disorders	9.50	1.67	14.69
6 Cardiovascular, Renal, Inflammatory, and Respiratory (*n* = 69,720: 7.61%)	**Chronic disease-drug group for atrial fibrillation**	**40.07**	**5.99**	**45.54**
	**Chronic disease-drug group for heart failure**	**42.57**	**5.70**	**43.38**
	**Chronic disease group for bradycardias and conduction diseases**	**11.50**	**4.14**	**31.48**
	**Chronic disease-drug group for inflammatory arthropathies**	**11.16**	**3.34**	**25.41**
	**Chronic disease-drug group for autoimmune diseases**	**7.07**	**3.25**	**24.69**
	**Chronic disease-drug group for anemia**	**17.61**	**3.00**	**22.81**
	**Chronic disease-drug group for chronic kidney diseases**	**31.24**	**2.96**	**22.52**
	**Chronic disease-drug group for COPD, emphysema, chronic bronchitis**	**20.53**	**2.56**	**19.49**
	**Chronic disease-drug group for ischemic heart disease**	**17.01**	**2.22**	**16.86**
	**Chronic disease-drug group for peripheral vascular disease**	**5.36**	**2.10**	**15.97**
	Chronic disease group for chronic pancreas, biliary tract, and gallbladder diseases	4.88	1.64	12.45
	Chronic disease-drug group for cerebrovascular diseases	10.29	1.59	12.10
	Chronic disease-drug group for colitis and related diseases	15.91	1.56	11.90
	Chronic disease-drug group for prostate diseases	14.98	1.48	11.27
	Chronic disease-drug group for hypertension	83.43	1.47	11.16
7 Multisystem (*n* = 49,592: 5.41%)	**Chronic disease group for other digestive diseases**	**23.29**	**9.70**	**52.46**
	**Chronic disease-drug group for dementia**	**21.63**	**6.48**	**35.07**
	**Chronic disease group for chronic pancreas, biliary tract, and gallbladder diseases**	**16.95**	**5.69**	**30.77**
	**Chronic disease-drug group for autoimmune diseases**	**10.95**	**5.03**	**27.19**
	**Chronic disease-drug group for inflammatory arthropathies**	**14.80**	**4.43**	**23.97**
	**Chronic disease-drug group for anemia**	**19.82**	**3.37**	**18.25**
	**Chronic disease-drug group for atrial fibrillation**	**14.80**	**2.21**	**11.96**
	**Chronic disease-drug group for heart failure**	**16.37**	**2.19**	**11.87**
	Chronic disease-drug group for colitis and related diseases	19.34	1.90	10.29
	Chronic disease-drug group for chronic kidney diseases	19.73	1.87	10.12
	Chronic disease group for bradycardias and conduction diseases	4.49	1.61	8.73
	Chronic disease-drug group for COPD, emphysema, chronic bronchitis	12.50	1.56	8.44
	Chronic disease-drug group for cerebrovascular diseases	10.06	1.55	8.41
	Chronic disease-drug group for esophagus, stomach, and duodenum diseases	9.95	1.37	7.41
	Chronic disease-drug group for osteoarthritis and other degenerative joint diseases	25.47	1.31	7.10

Categories highlighted in gray are chronic disease categories; those in white are chronic disease-drug categories. Categories in bold reach the O/E ratio threshold of two. Abbreviations: O: disease prevalence in the cluster; O/E ratio: observed/expected ratio; Ex: exclusivity; COPD: chronic obstructive pulmonary disease.

## Data Availability

The datasets are not available, since researchers signed an agreement with the Information System for the Development of Research in Primary Care (SIDIAP) concerning confidentiality and security of the dataset, which forbids providing data to third parties. The SIDIAP is subject to periodic audits.

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
