# Peer review of "Combined Multimorbidity and Polypharmacy Patterns in the Elderly: A Cross-Sectional Study in Primary Health Care"

_ijerph, 2021, doi:10.3390/ijerph18179216_

Round 1

Reviewer 1 Report

Combined multimorbidity and polypharmacy patterns in the elderly: a cross-sectional study in Primary Health Care

Thank you for your work on multimorbidity and polypharmacy, which are growing preoccupations in our aging societies. I must admit that the statistical methods used in this study are new to me, so that I consulted with the experienced statistician of our Centre. We discussed both the methods and results of the manuscript and want to point out several issues which need clarification.

  • Together we were wondering about the precise reasons these specific clustering techniques were chosen for this study. We would have liked to find a more detailed explanation of the choice of the methods.
  • More specifically, as the results of the process are described (page 6 of 24), the authors state that a 7-cluster model was identified, but we are not told how and why.
  • Also, there were some missing numbers in table A4 (after the comma) and I would like to see an explanation of the abbreviations used below the table.
  • On page 12 of 24, you say: “Additionally, this group includes those who have neoplastic diseases due to drugs that are neither indicated nor prescribed in primary healthcare. These data display once again that multimorbidity patterns depend on the diseases and drugs that are considered necessary for their composition [34].” Do you really mean that this group included patients having developed cancer as a result of using medications not usually prescribed in primary care? Or do you rather want to say that patients using medications against neoplastic diseases are also included in this group and that they can be identified by the use of medications not usually prescribed in primary care?
  • I think the results on the clusters resulting from the statistical analyses need to be described in more comprehensive way. I am unsure I understand whether clusters overlap or not? And how they compare to each other regarding the importance of possible adverse outcomes?
  • Finally, I do not understand (yet) how all this will help with clinical care or follow-up. Is there additional research underway linking the clusters to clinical outcomes, hospitalizations, mortality, specialist consultations etc.?
  • As is often the case with really new techniques, the use of this machine learning clustering technique needs a bit more explanation for the reader who is completely new to these research methods. The authors say in the introduction that analyses using data on both multimorbidity and polypharmacy may overestimate (the occurrence of adverse outcomes? ) since those two phenomena are highly associated, but they do not really explain how there method will change this. Is the idea that a patient, instead of being classified with a) morbidity and b) polypharmacy will just belong to c) a cluster, thus reducing the number or variables in an analysis using these characteristics? Or is there another aim to this research? I really think we need to be told a bit more here.

Reviewer 2 Report

Stafford and colleagues conducted a cross-sectional analysis to identify the combined multimorbidity and polypharmacy patterns in the elderly Catalan primary care population in Spain. The authors retrieved data from a very large, high-quality database (SIDIAP) containing electronic health records of a representative sample of patients. The study finding is of great interest and carries major implications for informing clinical practice guidelines on multimorbidity and polypharmacy.

Minor comments – 

1. Lines 28-31: I am a little bit confused about the way that the results are currently presented in the Abstract section. Does it mean that patients belonging to the "Diabetes" cluster had the highest proportion of multimorbidity and the highest proportion of polypharmacy at the same time, and that patients belonging to the "Cardiovascular, Renal, Inflammatory, and Respiratory" cluster had the lowest proportion of multimorbidity with polypharmacy? 

2. Line 32: the authors concluded that "there are clear trends in combined multimorbidity and polypharmacy patterns" in the Abstract section. However, the term "trend" was not described throughout the remaining part of the manuscript. 

3. Line 188: please be specific about which part(s) of the analysis were done by R, and Stata, respectively.

Reviewer 3 Report

63. the is the word “for” repeated twice

Results and Discussion

Personally, I have some concerns that Clusters 1 and 7 are discussed only in the Discussion, while they are not at all reported in the Results section. Consider the matter of method in reporting the results, and placing them in the correct section of the article. In the Discussion, the same issue (Cluster 1 and 7) can be taken up in different words, suitable for discussion.

I would also suggest putting the Table1 in the Results section.
